# Foundational Principles and Adaptation of the Healthy and Pathological Achilles Tendon in Response to Resistance Exercise: A Narrative Review and Clinical Implications

**DOI:** 10.3390/jcm11164722

**Published:** 2022-08-12

**Authors:** Kohle Merry, Christopher Napier, Charlie M. Waugh, Alex Scott

**Affiliations:** 1Department of Physical Therapy, University of British Columbia, Vancouver, BC V6T 1Z3, Canada; 2Centre for Hip Health and Mobility, Vancouver, BC V5Z 1M9, Canada

**Keywords:** exercise therapy, physical therapy modalities, rehabilitation, tendons, tendinopathy, mechanotransduction

## Abstract

Therapeutic exercise is widely considered a first line fundamental treatment option for managing tendinopathies. As the Achilles tendon is critical for locomotion, chronic Achilles tendinopathy can have a substantial impact on an individual’s ability to work and on their participation in physical activity or sport and overall quality of life. The recalcitrant nature of Achilles tendinopathy coupled with substantial variation in clinician-prescribed therapeutic exercises may contribute to suboptimal outcomes. Further, loading the Achilles tendon with sufficiently high loads to elicit positive tendon adaptation (and therefore promote symptom alleviation) is challenging, and few works have explored tissue loading optimization for individuals with tendinopathy. The mechanism of therapeutic benefit that exercise therapy exerts on Achilles tendinopathy is also a subject of ongoing debate. Resultingly, many factors that may contribute to an optimal therapeutic exercise protocol for Achilles tendinopathy are not well described. The aim of this narrative review is to explore the principles of tendon remodeling under resistance-based exercise in both healthy and pathologic tissues, and to review the biomechanical principles of Achilles tendon loading mechanics which may impact an optimized therapeutic exercise prescription for Achilles tendinopathy.

## 1. Introduction

Resistance-based therapeutic exercise is the cornerstone of non-surgical Achilles tendinopathy (AT) management [1,2]. Understanding how and why such exercises influence the experience of tendon pain and what factors may govern these effects may aid clinicians and researchers in optimizing therapeutic exercise interventions. Additionally, understanding the impact of therapeutic exercises on tendon function and the changes to the morphological, material, and mechanical properties of the tendon is critical for load management. Despite the prevalence of therapeutic exercise AT management, few works have explored tissue loading optimization for individuals with tendinopathy.

Although passive and relatively inelastic structures [3], tendons facilitate joint movement by transferring forces generated by muscles to the skeleton [4]. Specifically, tendons deform under load to store and return strain energy, making them critical during locomotion [5,6]. Human tendons vary considerably throughout the body in terms of structure [7] and mechanical properties [8], largely attributable to the functional demands of different regional loading environments [3,9,10]. The Achilles is the largest, strongest, and thickest tendon in the body [11], often experiencing forces of 5–7 bodyweights per step during running [12,13,14] and up to 7.3 bodyweights during single-leg hopping [15]. With repetitive or intense loading exceeding physiological limits, individuals may develop AT [16,17].

Achilles tendinopathy is defined as consistent pain in the Achilles tendon coupled with a loss of function associated with mechanical loading [18]. The incidence of AT is approximately 0.2–0.3% in adults (i.e., 2–3 per 1000) [19]. The occurrence substantially increases in runners, with incidences of 5.0–10.9% [20,21,22] in recreational runners and up to 52% in male former elite runners [23]. Achilles tendinopathy can either be classified as insertional AT (symptoms localized 0–2 cm from the distal insertion; 20–25% of Achilles tendon injuries) or midportion AT (symptoms localized 2–7 cm proximal to the insertion; 55–65% of Achilles tendon injuries) [1,24]. Diagnosis of insertional AT can often be confounded by additional pathologies, such as Haglund’s deformity, retrocalcaneal bursitis, and retrocalcaneal exostosis [25]. Given structural and functional differences across the Achilles tendon [26], it is important to distinguish between insertional and midportion AT as treatment option efficacy can differ [1,25]. Achilles tendinopathy can result in substantial localized pain and morphological changes to the tendon leading to deficiencies in material properties and mechanical behaviors [27]. If continuously subjected to the same detrimental loading patterns, the tendon structure can deteriorate further increasing the chance of rupture [28].

It is well established that resistance exercise positively remodels the healthy Achilles tendon [29,30,31,32]. Additionally, therapeutic exercise is consistently touted as a standard non-surgical treatment for AT [1,2], largely independent of muscle contraction type (i.e., concentric, eccentric, or isometric) [33,34,35,36]; however, the mechanism of therapeutic action is still a subject of debate and exploration [36,37,38,39]. Resultingly, much of the clinical research for AT has focused on combining resistance exercises with other treatments as opposed to optimizing the exercise program itself [40,41]. Although aspects of loading optimization have been investigated in healthy persons [42,43,44], the translation and applicability of these principles to individuals with AT has not been reported.

The purpose of this narrative review is to: (1) review the principles of tendon remodeling under resistance-exercise induced loading for both healthy and pathologic tissues; and (2) comment on the biomechanical principles of Achilles tendon loading mechanics, which may impact an optimized therapeutic exercise prescription for AT.

## 2. Anatomy Tailored for Function

This section provides a brief overview of several major anatomical considerations related to the biomechanics of the Achilles tendon and the triceps surae muscle-tendon unit (MTU).

### 2.1. Achilles Tendon Homeostasis and Structure

As outlined by Thorpe and Screen [3], the Achilles tendon is composed of approximately 20% cellular material and 80% extracellular matrix (ECM). Approximately 55–70% of the ECM is water, with the remaining portion corresponding primarily to highly organized Type I collagen and to a lesser extent Type III, V, and XI collagen, as well as non-collagenous molecules, such as proteoglycans, which promote ECM organization. The ECM is actively regulated by tendon fibroblasts, also known as tenocytes, which present with an elongated morphology and function primarily to control collagen synthesis. Importantly, tenocytes are mechanosensitive and have several force-sensitive provisions, such as integrins and stretch-activated ion channels, which allow them to modulate tendon collagen and non-collagenous content through cell signaling pathways thereby influencing tendon tissue mechanical properties [45]. Tenocytes are distributed both within and between tendon fascicles, which are a distinct unit amongst the tendon structural hierarchy. Amongst the collagen fibers, tenocytes form a three-dimensional network with cellular extensions expanding into the ECM [46], allowing them to sense substrate strain [47] and communicate these signals to adjacent cells via gap junctions [48] thereby promoting load monitoring throughout the tendon.

### 2.2. Force Transmission within the Achilles Tendon

Components of the tendon microstructure including collagen, elastin, and tenocytes are generally oriented along the longitudinal axis, resulting in anisotropic behavior and high tensile strength [49,50]. Additionally, the fluid within the tendon gives it viscoelastic properties [49,51]. The Achilles tendon is structured to temporarily store and return large amounts of kinetic energy from primarily tensile loads, some exceeding 9 kN [52,53], which is critical for efficient movement [4,49]. The Achilles tendon also optimizes the force generated by the triceps surae muscles by governing the force-length-velocity relationship [54,55]. On the proximal end, the Achilles tendon is the tendinous continuation of the triceps surae which proceeds to medially rotate until inserting distally on the posterior calcaneus [56,57]. As such, the proximal end of the Achilles tendon is cyclically deformed by the triceps surae muscles, while the distal end is fixed to the calcaneus via the enthesis, which serves to mitigate stresses at the hard-soft tissue interface [58]. The primary loading profile of the Achilles tendon underlines that stress and strain vary across the Achilles, but controlling tensile loading along the longitudinal axis is critical [59,60].

Tissue mechanics at all levels of the tendon hierarchy promote the load tolerance of the Achilles tendon [49]. Briefly, the smallest level of the hierarchy is the tropocollagen molecule, which is the structural unit of collagen fibers, and is composed of three polypeptides forming a triple-helix structure stabilized by hydrogen bonds [61]. Tropocollagen molecules are extensible under tensile loading via helix elongation [62] and lateral molecular order increases when tension is applied [63], possibly indicating alignment with the principal loading direction [49]. Staggered tropocollagen molecules self-assemble to form collagen fibrils [49], which take a mature form when covalently cross-linked through the enzymatic action of lysyl oxidase [64,65]. Cross-links are fundamental to the load-bearing capacity of the fibril [66,67], and cross-link density (or lack thereof) directly influences tendon mechanics by governing intra-fibril sliding [68]. At the fiber level, the collagen fibrils are oriented along the longitudinal axis in a distinct pattern known as ‘crimp’, which contributes to load tolerance as the crimp-pattern straightens near the onset of tensile loading [69]. Additionally, collagen fiber sliding appears crucial to tendon elongation [70]. Collagen fascicles are generally considered continuous throughout the tendon [49] and may act primarily as independent load-bearing structures with negligible lateral force transmission at low strain levels [71]. With that said, work investigating mechanical loading at higher load levels (up to the point of rupture) concluded that both the spiral twisting of the fascicles and sliding within the Achilles tendon considerably improve tissue strength by more evenly distributing stresses across the whole tendon [72]. In sum, features throughout the tendon hierarchy are responsible for global tendon elongation, though testing heterogeneity makes it challenging to isolate relative contributions [49].

### 2.3. Force Transmission within the Triceps Surae Muscle-Tendon Unit

The triceps surae (i.e., medial gastrocnemius [MG], lateral gastrocnemius [LG], and the soleus [SOL]) is responsible for the majority of plantarflexion force generation which enables locomotion [73,74]. While the uniarticular SOL acts as the main plantar flexor muscle [75], the bi-articular gastrocnemius functions to both flex the knee and contributes to ankle plantarflexion [73]. In addition to function, SOL also differs from gastrocnemii in fiber type [76] and architecture [77,78]. The approximate 6:2:1 physiological cross-sectional area (CSA) relationship of the SOL, MG, and LG [79] indicates that the maximal force-production capacity of the SOL is considerably greater than that of the gastrocnemii as physiological CSA is directly linked to muscle force production [80,81]. Each of the triceps surae muscles insert onto the calcaneus by way of three different ‘subtendons’, which originate from each muscle and represent distinct functional portions of the Achilles tendon [56,57] (Figure 1). Both the subtendons and the fascicles comprising them rotate counterclockwise in the right limb and clockwise on the left, though the extent of rotation varies considerably between individuals [56,82,83]. Further, the fascicles tend to fuse distally creating a more uniform tendon structure [72,82]. The difference in muscular force potentials and subtendon transmission pathway through the Achilles may have implications on the incidence of AT through modifications to tendon mechanical properties [84], strain distribution [60,85,86], and shear generated from subtendon sliding [26,86,87].

## 3. Tendon Tissue Remodeling

Despite the complex loading mechanics of the triceps surae MTU, not all loading is detrimental to tendon health. While extrinsic factors contributing to tendon damage appear to be primarily attributable to submaximal cyclic loading, such as those induced by running and other training-related factors [88], targeted tendon loading of adequate magnitude can induce positive changes in tendon morphological, material, and mechanical properties [29,32]. Specifically, mechanotransduction details the body’s ability to translate mechanical loading into structural tissue change via cellular responses [89].

### 3.1. Healthy Tissue Remodeling

Mechanosensitive cells are responsive to tension, compression, and shear [90]. Loading magnitude [29,32], and perhaps more precisely strain [42,43,44], appears to modulate mechanotransduction in the healthy Achilles tendon. Specifically, strain magnitude, frequency, rate, and duration influence tenocyte biochemical processes [91,92,93] and gene expression [94,95]. For adequately long intervention durations (generally 12 weeks [36]) loads of greater than 70% of maximum voluntary contraction (MVC) [29,32] or strains of 4.5–6.5% [42,43,44] may deliver the appropriate loading-induced tendon stimulus to initiate mechanotransduction pathways; however, the relationship of tendon force and resulting strain can vary substantially between individuals [96,97]. Additionally, strain calculated as the displacement of the gastrocnemius medialis myotendinous junction from its resting length may differ from strain calculated as the change in length of the free tendon, which is more compliant [98,99], and perhaps where the majority of strain occurs. Theoretically, only looking at strain across the free tendon could change the ‘optimal’ adaptation threshold of 4.5–6.5% strain [42,43,44] typically arising from loading programs of greater than 70% of MVC [29,32].

Although the metabolic activity of tendon is low and the structure is typically static, loading-induced stimuli may trigger mechanotransduction and anabolic signaling pathways in the tendon [3]. In particular, the upregulation of insulin-like growth factor (IGF-I), among other growth factors, influences cellular proliferation and matrix remodeling [89,100,101]. Positive matrix remodeling appears to be largely attributable to a net synthesis of type I collagen, thereby making the tendon more load-resistant, though components of the ECM—proteoglycans, glycosaminoglycans, and cross-links—are also influenced by mechanical loading and contribute to macroscopic tendon behaviour through their actions on collagen fibrils [49,100]. Mechanically, longitudinal stiffness (resistance to deformation) increases [29,32,102], and strain for a given tendon force decreases [43,103] in response to increased loading in vivo. Material properties increasing in response to increased loading in vivo include modulus [29,32,102]. Morphologically, tendon CSA increases in response to increased loading in vivo [29,32,102], though limited evidence suggests that transient fluid redistribution may mask this in the short-term [51,104]. Additionally, loading-induced changes may differ along the Achilles tendon as the regional variation in load management [98,105,106] may preferentially activate mechanotransductive pathways leading to region-specific tendon hypertrophy [43,44]. Though still an area of exploration, the opposite could also be the case in that the non-uniform stress distribution within the Achilles tendon could contribute to the location of abnormalities associated with AT [107]. Moreover, while the tendon changes/adaptations described above are primarily related to resistance training, it appears that other types of mechanical loading, such as cyclic loading (e.g., running), can also induce adaptation in the healthy Achilles tendon [108,109]; however, conflicting evidence suggests that some other types of mechanical loading, such as plyometric exercises, may or may not adapt the Achilles tendon in a similar fashion [110,111,112,113,114,115].

### 3.2. Pathologic Tissue Remodeling

The pathogenesis of tendinopathy appears multifaceted, which has given rise to various pathophysiological theories [36]. Current rhetoric suggests that initial cyclic overloading of the tendon leads to degeneration and disorganization of healthy collagen, which triggers an acute inflammatory response [36,87,101]. If the cyclic overloading is continued without intervention, the tendon pathology worsens through a positive feedback loop of injury to both the original and poor-quality repair tissue, inflammation, and failed repair. Macroscopically, evidence suggests that AT increases tendon CSA [116,117,118] and longitudinal strain [116,117,118], and decreases modulus [116,119], transverse strain [120], longitudinal stiffness [116,118,119], and transverse stiffness [121] in vivo. Taken together, these changes lead to functional deficits across the strength spectrum potentially increasing risk of AT recurrence [122,123,124].

Therapeutic exercise remains one of if not the most effective non-surgical approach for managing AT [1,2]. The suggested mechanism of action is generally considered to be restoration of tendon material, mechanical, and morphological properties similarly to healthy tendon remodeling [36,37,41], thereby improving functional strength [33]. Macroscopically, evidence suggests that targeted mechanical loading decreases tendon thickness [125] and volume [126]; however, there is a paucity of evidence underpinning the restoration of tendinopathic tissue capacity, with most studies focusing on functional and acute analgesic effects [36]. Evidence suggests that abnormal structure (i.e., hypoechoic areas and irregular structure) may normalize in some individuals following a 12-week eccentric exercise protocol [125,127], though the time needed for such changes to occur may vary [38]. Additionally, Cook and colleagues [128] posit that exercise-based adaptation may build capacity in the area of aligned fibrillar structure instead of acting on the area of abnormal structure. Nonetheless, evidence suggests that structural changes do not entirely explain clinical outcomes [129,130]. Building on this idea, O’Neill, Watson, and Barry [37] highlight that tendon structure is not observed to significantly change over the typical intervention period. The authors further suggest that changes in neuromuscular output may explain clinical benefit, and that training should focus on increasing stiffness of the triceps surae MTU, increasing strength, and shifting the length-tension curve of the triceps surae muscles through sarcomerogenesis. Although still an area of exploration, it appears that therapeutic exercise for AT should focus on improving the mechanical and material properties of the entire MTU thereby simultaneously building strength capacity and neuromuscular control [131].

## 4. Biomechanical Considerations towards Optimal Exercise Prescription

The optimal therapeutic exercise intervention for tendinopathy is unknown [41,132]. This may be in part due to the substantial number of tunable parameters which comprise an exercise prescription [133], or the heterogeneity and underreporting of resistance training features for tendinopathy [40]. As such, the authors have chosen to report on several fundamental exercise parameters for managing AT, including muscle contraction type; load intensity; loading frequency, rate, and duration; exercise positioning; and the exercise schedule.

### 4.1. Muscle Contraction Type

*Contraction type* describes the change in length of a muscle during a contraction. While isometric contractions generate force without changing the length of the muscle, isotonic contractions generate force as the muscle either lengthens (i.e., eccentric contraction) or shortens (i.e., concentric contraction). Seminal therapeutic exercise protocols for tendinopathy have popularized the use of eccentric training [134,135], although several mixed exercise protocols (i.e., those incorporating both the eccentric and concentric phases of a movement) [136,137] have also demonstrated comparable results [34,35] drawing into question the role of muscle contraction type in treating AT. Additionally, isometric exercise protocols may be a viable option for treating tendinopathies [138], though conflicting evidence exists specifically for their use in managing AT [139,140]. With the mechanism of therapeutic action of mechanical loading on AT still unresolved [36,37,38,39] and given a lack of evidence favoring one contraction type over another [141,142,143], it does not seem justified to exclusively prescribe eccentric exercises for AT.

As opposed to universally prescribing eccentric exercises for tendinopathy, Millar et al. [36] suggest that clinicians should instead focus on conveying the principles of tendon loading and individually tailor the program to promote patient engagement with and consequently the success of the program. Importantly, individuals with AT have highlighted the burden of therapeutic exercise, particularly those prescriptions incorporating daily or twice daily exercises [144]. With the biopsychosocial impact being a core tenet of AT [144,145,146], and coupled with patient perceptions that passive treatments (e.g., massage, dry needling, ultrasound) are more efficacious than therapeutic exercises for managing AT [145], specifying exercises based on muscle contraction type may be less important than patient education about mechanical loading and the prescription of exercises according to what the patient is most likely to perform [36]. Adherence rates and patient satisfaction with mixed protocols appear to be at least as good if not slightly better than purely eccentric protocols [34,35], though more comparative studies are needed to confirm this assertion.

### 4.2. Load Intensity

*Load intensity* describes the magnitude of the training stimulus, or put differently, the amount of resistance applied. In healthy persons, load intensities of greater than 70% of MVC significantly induced tendon stiffness adaptation regardless of contraction type [29,32]. Further, if the optimal range of tendon strain causing adaptation is considered (4.5–6.5%), it appears that 90% of MVC may be more appropriate [42,43,44]. However, these studies favour a young, healthy male population, limiting generalizability. Work in our group with healthy individuals has also highlighted possible issues pertaining to participant tolerability when exercising consistently at 90% of MVC [103]. A recent systematic review and meta-analysis by Lazarczuk et al. [32] found that resistance training protocols completed by healthy individuals at high-intensities (greater than 70% of 1RM/MVC) elicited large increases in tendon stiffness and modulus, and small increases in CSA. In contrast, low-intensity protocols (less than 70% of 1RM/MVC) produced moderate increases in tendon stiffness, large increases in modulus, and no clear change in CSA. The authors also found that high-strain protocols (~5%) elicited significantly larger increases in tendon stiffness and modulus when compared to low-strain protocols (~3%).

Individual variability also plays an important role when prescribing load intensities to elicit specific strain values. In athletes, Achilles tendon strain during MVC tests vary substantially [96], highlighting the challenge in specifying load-intensity without knowing first how it correlates to tendon strain in the individual. Similarly, in a group of 20 healthy adults (10 male/10 female; mean age 25.7 ± 2.9 yrs; mass 70.0 ± 10.8 kg; height 170.0 ± 8.4 cm), tendon strain levels were found to vary considerably with the percentage of MVC (Figure 2). Given the 4.5–6.5% optimal strain adaptation threshold [42,43,44], these results demonstrate that anywhere from 30% to 90% of MVC may elicit these strain levels depending on the individual; however, staying above 60% of MVC appears to achieve the target strain threshold in two-thirds of (healthy) individuals in this small sample. When coupled with the above review papers highlighting the importance of high strain levels for tendon adaptation [29,32], a threshold of greater than 70% of MVC appears practical for achieving high tendon strain in most healthy individuals.

Translating such evidence to individuals with AT is challenging for several reasons. Firstly, given that AT decreases longitudinal strain [116,117,118] and plantar flexor strength [122], it is unclear if those with AT can: (1) consistently reach the optimal strain range; (2) consistently tolerate loads of 70–90% of MVC; and (3) if the optimal strain range causing adaptation applies for tendinopathic tissue. Secondly, MVC testing is not always practical clinically, and home-based approaches for monitoring rehabilitative loading according to MVC are non-existent. A close comparative methodology currently being tested is heavy slow resistance training (HSR), which functions off of a ‘repetitions maximum’ (RM) prescription approach [137]. HSR, which maximally reaches six RM, has been adapted for AT and has demonstrated clinical benefits while also being well received by patients [35]. However, more research for HSR is required [142], and adherence and fidelity to exercise dose during HSR in individuals with AT may pose an issue [148]. Additionally, the importance of load progression for AT rehabilitation cannot be overstated [1,2,41,149]. Without sufficient forethought into both exercise individualization and progression, it is perhaps easy to miss the ‘sweet spot’ of tendon training clinically resulting in a plateau effect of rehabilitation [97].

### 4.3. Loading Frequency, Rate, and Duration

*Load frequency* describes the number of complete repetitions (loading and relaxation) that can be completed over a specific time. Load frequency is calculated as the inverse of the total time needed to complete a single repetition, and is measured in hertz (Hz). *Loading rate* describes the change in loading intensity with respect to time (e.g., time taken to reach target load from no-load condition). *Load duration* describes the amount of time a training stimulus is applied for.

In healthy individuals, Arampatzis, Bohm, and colleagues demonstrated that when a high load-intensity is used (i.e., 90% of MVC), a low loading frequency (i.e., 0.17 Hz, 3 s loading/3 s relaxation) was superior when compared to a high loading frequency (i.e., 0.5 Hz, 1 s loading/1 s relaxation) [42], whereas the high loading rate (i.e., one-legged jumps) and high loading duration (i.e., a single 12 s isometric plantar flexion contraction per set) yielded inferior adaptive results compared to the reference protocol (4 × 6 s isometric contractions at 90% of MVC) [44]. Beyond the Achilles, it appears that conflicting evidence exists as to the effect of loading rate [150] and duration [151,152] on tendon adaptation. Given the impact of rest duration on collagen organization [103], more research into this factor may be warranted.

There exist many therapeutic exercise protocols for AT; however, there are four which are consistently cited within the literature [33]—HSR [137], Alfredson’s eccentric [135], Silbernagel’s combined [136], and Stanish and Curwin’s eccentric-concentric protocol [134]. Of the original published protocols, HSR was the only protocol to explicitly state load frequency, citing 6 s/repetition. The others opted to instead use adjectives (e.g., ‘slow’, ‘moderate’, ‘fast’) to specify load frequency as these are practical for home-based rehabilitation. Consequently, much of the AT rehabilitation evidence to date largely ignores the effects of loading frequency, rate, and duration generally prioritizing clinical outcomes (e.g., pain, function) [40].

### 4.4. Exercise Positioning

*Exercise positioning* describes the orientation of the body in space when completing a therapeutic exercise protocol. Based on current understanding, optimal positioning for therapeutic exercise of the Achilles tendon must: (1) facilitate the generation of controlled high-magnitude loading through the triceps surae MTU; (2) be tolerable and practical to the client within the context of executing an exercise prescription in said position(s); and (3) be repeatable both during exercise completion and when taking measurements used for outcome measures. Knowing these constraints, there are two main considerations: lower limb joint angles (i.e., ankle, knee, hip) and weight-bearing (WB) versus non-weight-bearing (NWB).

Mechanically, load through the triceps surae MTU is dependent on its distal anchor at the calcaneus and its proximal anchors on the tibia and fibula (soleus) and medial and lateral condyles of the femur (gastrocnemii). Because the soleus does not cross the knee joint, its force output and EMG activity are independent of knee angle [153,154,155]. In contrast, at shorter muscle lengths, such as high degrees of knee flexion the gastrocnemius muscle fascicles are de-recruited and shorten [155,156,157], which lessens its force-generating ability and EMG activity [153,154,155,158]. Resultingly, less force is transmitted through the Achilles tendon during knee flexion both passively [159] and actively [155,160], though contradictory evidence exists [161]. Plantarflexion torque [86,158] and Achilles tendon displacement [162,163] appear greater in knee extension and dorsiflexion, and maximum dorsiflexion angle predicts force through the Achilles tendon [161]. Importantly, knee extension [164] and hip flexion [165] both decrease maximum dorsiflexion angle, and the combination of both limits ankle range of motion [166]. Although no muscle spans the entire lower limb from the hip joint to the ankle, neural tension from the sciatic nerve appears to be primarily responsible for limiting ankle range of motion [167]. With that said, knee flexion above 20 degrees appears to eliminate the dorsiflexion restraining effect of the gastrocnemius [164]. Taken together, most evidence suggests that despite the dorsiflexion restraining effect of knee extension, the force through the Achilles tendon and subsequently the Achilles displacement are superior in knee extension. Furthermore, such research in healthy individuals suggests that greater ankle dorsiflexion, knee extension, and indirectly hip extension may position the body to generate maximal plantar flexor torque thereby maximally straining the Achilles tendon. Of these lower limb joint angles, ankle angle appears to be most deterministic of Achilles tendon loading as ankle angle largely dictates the force through the Achilles tendon [161] and tendon elongation [168]. Although promising for midportion AT, a caveat exists for individuals with insertional AT where loading in dorsiflexion may be irritable and should be avoided, at least in the early stages of rehabilitation [41].

Standing calf raises and heel drops align with this positioning. Building upon this, WB significantly enhances ankle dorsiflexion compared to NWB across knee angles in healthy individuals [164]. Because of the greater dorsiflexion angle and the effect of body weight, peak Achilles tendon loading is significantly higher when WB [161]. However, the HSR protocol [137] tested in Yeh et al.’s work [161], uses a knee-flexed sitting position as opposed to long-sitting where the knee is extended; therefore, more research is needed to discern whether the WB status itself, the knee extension when WB, or both, is driving the peak Achilles tendon loading. Practically speaking, it seems that WB is preferred in AT rehabilitation protocols [33] because of its ability to deliver sufficient, reproduceable loads to the Achilles tendon, and has less to do with potentially optimal joint angles.

With strain being a driver of mechanotransduction [42,43,44] and deficits in plantar flexor strength being a primary biomechanical feature of AT [122,123,124], one can speculate that this positioning regime (ankle dorsiflexion, knee/hip extension) may be most appropriate for AT therapeutic exercise. Despite a dearth of evidence pertaining to the manipulation of joint angles in AT rehabilitation, Reid et al. [154] found that gastrocnemius EMG activity was most active in knee extension, and soleus activity was constant between knee flexion and extension. The authors further suggest that the WB bent-knee condition specified in Alfredson’s original protocol (angle unspecified) [135] may therefore be unnecessary for maximizing soleus activation. However, for a given applied load (e.g., when WB), a bent-knee position would disadvantage the gastrocnemius thereby increasing loading of the soleus, which may be therapeutically important. Additionally, changing the ankle or knee angle or the contraction type (i.e., passive rotation, isometric plantarflexion, or eccentric contraction) also appears to impact regional (i.e., deep or superficial) Achilles tendon tissue displacements [85,162,169,170,171], though applicability is limited by small samples of primarily asymptomatic individuals. Further, contradictory evidence suggests that knee angle may not play a role in Achilles tendon tissue displacement patterns [172]. Expanding upon the idea that a bent-knee position targets the soleus, the HSR program [137] calls for a third of the program to be completed at 90 degrees of knee flexion. With that said, Alfredson’s protocol and HSR have both demonstrated substantial clinical benefits [35] perhaps questioning the need to prioritize knee extension or joint angles at all. A significant amount of research appears to be needed before ‘optimal’ joint positions for AT therapeutic exercise can be empirically identified. With a lack of research into mechanotransduction of tendinopathic tissue and joint angle manipulation within AT populations, caution must be advised when interpreting these suggestions.

Some attention may also be paid to the practical considerations of symptomatic individuals being able to execute these therapeutic exercises at home. Generally, exercise therapy for AT has favored WB exercises with a fully extended hip and knee (e.g., calf raises, heel drops) [33]. WB positions for home-based therapies are preferable for AT rehabilitation as they facilitate high-magnitude loading of the Achilles by forcing the tendon to counteract the weight of the body. Additionally, WB positions require little set up or equipment which may promote adherence to the exercise prescription. Although home-based seated exercises for AT are common (e.g., strengthening exercises using elastic resistance-based equipment, such as the TheraBand^TM^ [Performance Health, Akron, OH, USA]), these exercises can be limited by their capacity to apply high-magnitude resistance to stimulate adaptation of the triceps surae and Achilles tendon. Gym-based options are also available in both the WB and NWB positions (e.g., seated calf extension, seated or standing calf raise, seated or standing smith machine variations). Despite gym-based exercises having the ability to facilitate high-magnitude loading, many individuals consider these options less accessible than home-based options. One must also consider that some populations may have difficulty getting into or sustaining certain positions. For example, individuals with increased dural tension may not tolerate long sitting. In sum, although research in healthy individuals suggests that greater ankle dorsiflexion, knee extension, and hip extension may facilitate high-magnitude loading of the Achilles tendon thereby stimulating positive adaptation, clinical expertise must couple these potential insights with client-specific factors towards generating an individualized and progressive exercise therapy protocol for AT.

### 4.5. Exercise Schedule

The *exercise schedule* describes the frequency of exercise in terms of the intervention duration, number of therapeutic exercise sessions per week or per day, and number of sets/repetitions to be completed per session. A recent systematic review by Burton and McCormack [40] synthesized the resistance training protocols amongst exercise interventions treating lower limb tendinopathies; Achilles tendinopathy was the most commonly investigated within this review (26/52 randomized controlled trials [RCTs] included), followed by patellar tendinopathy (16/52 studies included). Therapeutic exercise interventions ranged in duration from 4 to 26 weeks, with 85% of included studies using a 12-week intervention. Exercise session frequency ranged from two to seven days per week, and two to 14 exercise sessions per week. Exercise sets ranged from 1 to 12, and repetitions ranged from 3 to 30. The most commonly used therapeutic exercise program was the original (or slightly modified) ‘Alfredson’ protocol [135], which was cited in 48% of included studies. The Alfredson protocol prescription consists of 12 weeks of exercise sessions completed twice a day, seven days per week. Each session consists of two exercises (straight-leg and bent-knee eccentric heel drops), completed in 3 sets of 15 repetitions totaling 180 repetitions per day. Despite the frequent use of the Alfredson protocol, the need for the high volume of exercises suggested by this protocol has been drawn into question, with one study finding equal improvement at six weeks between a ‘standard’ Alfredson prescription and a ‘do-as-tolerated’ prescription, which completed the same exercise intervention excluding the repetition volume [173]. Within the study, the do-as-tolerated group averaged 112 repetitions per day, whereas the standard Alfredson group averaged 166 repetitions per day. On the whole, Burton and McCormack’s review suggests that substantial variability exists in exercise intervention programming for AT, with most protocols yielding positive therapeutic results. The authors go on to question the high-volume Alfredson protocol and suggest that high-magnitude progressive tendon loading with adequate rest periods may optimize tendon adaptation and subsequently improve clinical outcomes. However, with much of the literature continuing to use derivatives of the Alfredson protocol, it is challenging to identify what constitutes an optimal exercise schedule for managing AT until more research emerges using lower volume, high-magnitude loading.

A summary table of the considerations detailed in Section 4.1, Section 4.2, Section 4.3, Section 4.4 and Section 4.5 can be found below (Table 1).

## 5. Limitations and Future Directions

Several important limitations of this commentary must be raised. Firstly, samples within the included literature were generally small and consisted primarily of young, healthy males, limiting generalizability. Secondly, only uniplanar motion was considered when multiplanar, progressive tendon rehabilitation is important for improving functional strength during AT rehabilitation [41]. A return-to-sport phase emphasizing functional movements and task specificity should also be used following the initial strength-focused ‘rebuilding’ phase of the rehabilitation protocol [41]. Pain processing, symptom alleviation, and psychosocial and contextual effects were also outside the scope of this commentary but are critical features of AT rehabilitation [41]. Additionally, individual clinical presentation, response to rehabilitation, and patient values must be considered at the forefront of AT rehabilitation. Evidence has consistently demonstrated that therapeutic exercise is a robust treatment strategy for AT, but further research is needed before optimal programming can be recommended.

Along with replicating some of the work done in healthy persons in those with AT, several other areas of exploration may progress therapeutic exercise for AT. Further exploration of the role of the Achilles subtendons, subtendon interfaces, and subtendon biomechanics is warranted [106]. Study of neuromuscular changes, pain processing, and psychological effects during rehabilitation may help illuminate the mechanism of action of therapeutic exercise [37]. Measuring Achilles tendon elongation during plantarflexion could promote individualization of exercise programming, particularly if paired with a practical system for quantifying MVC [96]. As further evidence emerges pertaining to the pathophysiology and adaptation of tendinopathic tissues, computational models may enable load-based testing with tendinopathic tissues without placing already compromised individuals at further risk [106,174]. By better understanding tendinopathic tissue adaptation and the biomechanical considerations underpinning therapeutic exercise for AT, clinicians may prescribe more targeted programs thereby reducing client burden and improving rehabilitation outcomes.

## 6. Conclusions

Resistance-based exercise therapy is one of the most prevalent management strategies for chronic AT. High-magnitude loading of the Achilles tendon can elicit positive tendon adaptation in both healthy and tendinopathic tissues thereby allowing the tendon to better tolerate load. Research in healthy tendons suggests that sufficiently high-magnitude loading is critical for tendon adaptation, though this research is not well studied within a tendinopathic population and may be constrained by client tolerability. Other factors which comprise a therapeutic exercise protocol (e.g., loading frequency, rate, duration) appear to be less deterministic of tendon adaptation in a healthy population and are again insufficiently described in a tendinopathic population. Greater ankle dorsiflexion, knee extension, and hip extension seem to position the lower body optimally to maximize Achilles tendon load, though more research is needed to discern whether an optimal position is necessary if high-magnitude tendon loading can be achieved through alternative strategies (e.g., use of additional resistance).

When treating AT using exercise therapy, clinicians should prioritize high-magnitude, repeatable Achilles tendon loading whether that be through WB exercises or NWB exercises incorporating additional resistance. Clinicians should focus on client tolerability and exercise repeatability as these factors may contribute to adherence rates. Although a variety of potential factors may be important for treating AT using therapeutic exercise (e.g., loading variables, positioning), the research in healthy persons must be replicated in tendinopathic individuals before further clinical suggestions can be made.

## Figures and Tables

**Figure 1 jcm-11-04722-f001:**
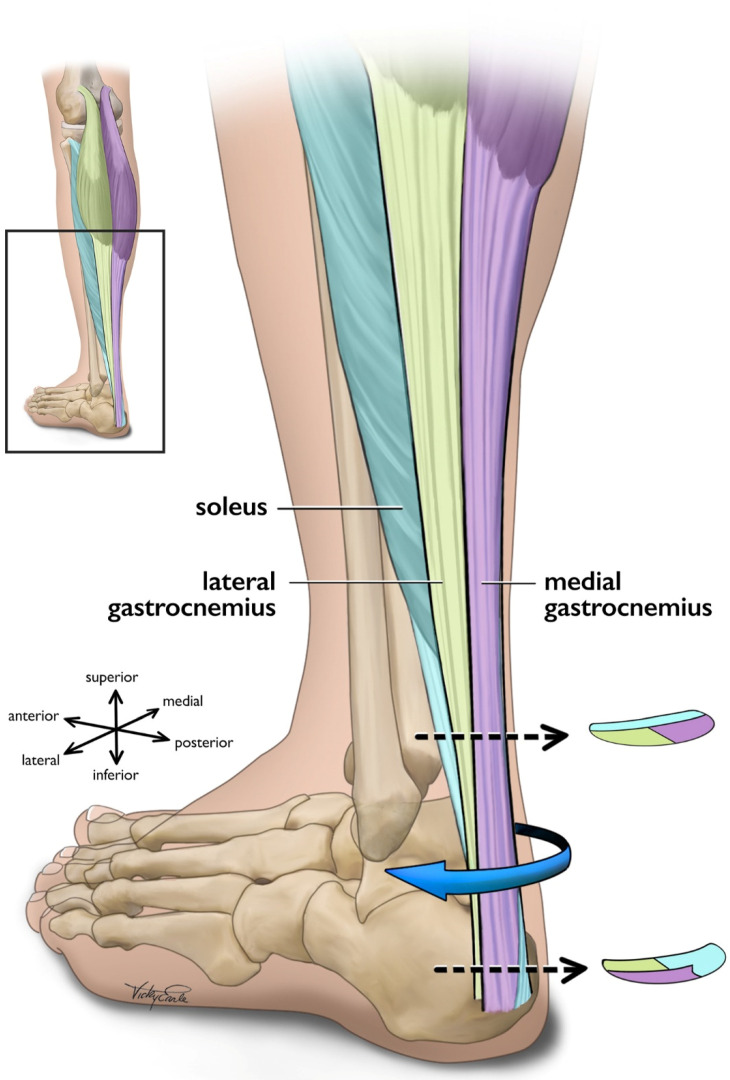
Posterolateral view of the left Achilles tendon and the three subtendons which comprise it. Subtendons rotate in a clockwise fashion traveling distally down the tendon. Cross-sectional views are displayed near the proximal and distal ends of the free tendon, and are based on cadaveric studies [56,82]. The soleus and soleus subtendon are colored teal, the lateral gastrocnemius and associated subtendon chartreuse, and the medial gastroc and its subtendon lavender.

**Figure 2 jcm-11-04722-f002:**
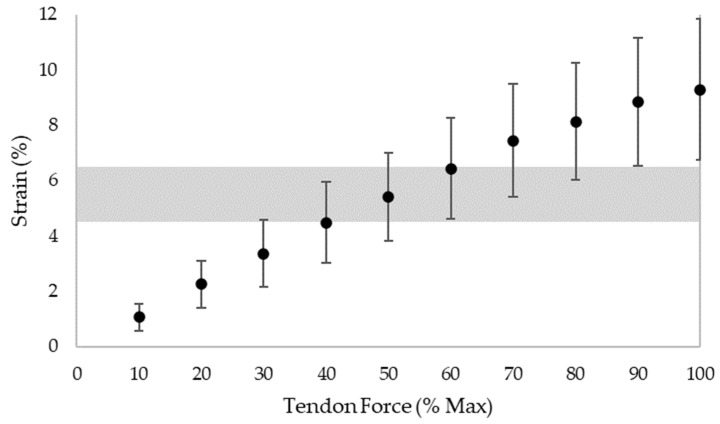
Relationship between tendon force (i.e., % MVC) and strain in 20 healthy individuals. The transparent gray area indicates the strain range proposed to be optimal for positive tendon adaptation (4.5% to 6.5% strain [42,43,44]). Data points represent means and error bars denote standard deviation. Data in this graph was obtained from [147].

**Table 1 jcm-11-04722-t001:** Clinical takeaways for resistance-exercise based Achilles tendinopathy management.

Biomechanical Consideration	Section	Summary Points	Clinical Recommendation
Muscle Contraction Type	Section 4.1	With a lack of evidence favoring one contraction type [141,142,143], it does not seem justified to exclusively prescribe eccentric exercises	- Different contraction types can be used to treat AT ^a^ - Focus on conveying the principles of tendon loading
Load Intensity	Section 4.2	High-magnitude loading (>70% of MVC ^b^) induces greater tendon adaptation in healthy individuals [29,32] Many AT exercise programs favor bodyweight loading and increase resistance as tolerated (e.g., 5 kg increments in a backpack) [40]	- Increasing load intensity appears to stimulate greater tendon adaptation in healthy individuals - Prioritize high-magnitude loading (as tolerated) and load progression over time
Loading Frequency, Rate, and Duration	Section 4.3	Evidence pertaining to these factors is limited Seminal AT rehabilitation programs prioritize ‘slow’ loading [134,135,136]	- Not enough existing evidence, though most programs use ‘slow’ loading frequencies
Exercise Positioning	Section 4.4	Of the lower limb joint angles, ankle angle appears to most impact Achilles tendon loading as it largely dictates the force through the Achilles tendon [161] and tendon elongation [168] WB ^c^ enhances ankle dorsiflexion compared to NWB ^d^ across knee angles in healthy individuals [164] Soleus activity is independent of knee angle [153,154,155]; gastrocnemius is less active in knee flexion [153,154,155,158]	- Ankle dorsiflexion, knee/hip extension may be most appropriate for AT therapeutic exercise - Excessive dorsiflexion may be irritable to those with insertional AT, and should be avoided, at least in the early stages of rehabilitation - WB positions are widely used within AT rehabilitation, but this may be because WB helps facilitate high-magnitude loading - Loading magnitude should be prioritized over exercise positioning
Exercise Schedule	Section 4.5	Most studies use 12-week long exercise interventions, though positive results have been found at six weeks [40] Of 52 RCTs ^e^, session frequency ranged from two to seven days per week, and two to 14 exercise sessions per week [40] Of 52 RCTs, sets ranged from 1 to 12, and repetitions ranged from three to 30 [40]	- A 12-week exercise program duration appears most appropriate - Exercise session frequency can vary considerably, and it largely depends on the loading intensity, volume, and tolerability - Sets/repetitions can vary considerably, and they largely depend on the loading intensity and tolerability

^a^ AT = Achilles Tendinopathy; ^b^ MVC = Maximum Voluntary Contraction; ^c^ WB = Weight-bearing; ^d^ NWB = Non-weight-bearing; ^e^ RCTs = Randomized controlled trials.

## Data Availability

Not applicable.

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
