# Peer review of "Foundational Principles and Adaptation of the Healthy and Pathological Achilles Tendon in Response to Resistance Exercise: A Narrative Review and Clinical Implications"

_jcm, 2022, doi:10.3390/jcm11164722_

Round 1

Reviewer 1 Report

This is a well written manuscript concerning the anatomy and physiology of the achilles tendon. Concerning the effect of exercise on AT, the manuscript raises more questions than answers. 

The manuscript describes the normal tendon reaction to exercise. 

The authors describe overload as a way to stimulate tendon growth. Normaly clinicians use the terms "overload" and "overuse" as too much, too many reps, too often to describe the cause of injury. I would choose another wording. When you have an injury, too much should primarily bee treated by less load, less reps. I think this loadreduction is the most important first step treatment, not mentioned in this manuscript.

Concerning strengthening exercises, many different exercises seems to work equally good or bad. Do they work at all, or is it the load control that is the important factor. One would need a "wait and see" control group, which has been done in different tendons, demonstrating no long term benefit of strengthening exercises in per ex tennis elbow and gluteus medius tendinopati.

How many patients with AT are actually satisfied with the exercise treatment. Ram in Canada and Wetke in Denmark have demonstrated that only 10-24% of all patients refered to doctors with AT are satisfied with exercise treatment after 3-6 months.

Another kind of exercise is stretching proven with equal effect as strength training in AT in some clinical studies. I would have liked a discussion of how fibroblast reacts if they are to be  pulled by a muscle ((strengthening exercise) or be pulled by a bone (stretching exercise). In vitro, it is the pulling per se that stimulates collagen synteses.

Author Response

Please see attached response to reviewers document. Thank you for your time!

Reviewer 2 Report

General comment

I wish to thank the authors for this work that can be an helpful tool for clinicians in order to guide their practice. The paper is well written and structured in an easy and very readable way.

As a suggestion in order to facilitate once more clinicians in the application of the information highlighted by the paper, I’d like to see a summary table at the end of paragraph 4. A kind of resume of the paragraph's content (4.1, 4.2, 4.3, 4.4) and a clinical suggestion (e.g. the type of contraction to use; the best positioning; etc.)

A piece of information is missing in my point of view, what does the literature say about the frequency of exercise in terms of how many exercise sessions in a week or per day.

Minor revision

Line 88 the word tendon is missing

Author Response

(The authors gave the same response as above.)

Reviewer 3 Report

This is a well-written narrative review summarising the latest research on Achilles tendinopathy. Nonetheless, I feel that large parts of this manuscript have been present in many narrative reviews and meta-analyses on tendinopathy in recent years. From my point of view, the principal aim of tissue loading optimisation for individuals with tendinopathy (Line 55ff in the Introduction) and thus the real added value of this review should be pointed out more clearly. I would also suggest addressing the differences between mid-portion or insertional tendinopathy and the related structure-function relationships or lack thereof a little bit more. Vice versa, I suggest referencing several excellent papers summarising, e.g., tendon anatomy and shortening some paragraphs.

Minor comments:

Title: I am not convinced by the title. It is unusual to speak from tendon performance, and it should contain that the manuscript is also (or mainly) related to pathological tissue.

Line 28f: We do not only want to know the influence on tendon pain but also function and maybe tendon properties (as mentioned later in the manuscript).

Line 41f: In terms of consistency, I suggest presenting the overall incidence of AT in adults also in per cent.

Line 44ff: The paper by Neal Miller and colleagues is not the best to report on morphological, material or mechanical deficiencies in tendinopathy. The authors should quote, e.g., Obst et al. Sports Med 2000 instead.

Subheading 2.1. The following parage is not only about the tissue structure but also about tissue maintenance of homeostasis etc., and should be chosen more precisely.

Line 176ff: The reported changes/adaptations of the tendon to mechanical loading are mainly related to resistance training. It also seems to be the case that cyclic loading (e.g., runner) induces similar patellar and Achilles tendon adaptations (Wiesinger et al., PlosOne 2016, Front Physiol 2017), but not necessary plyometric mechanical loading (e.g., Kubo et al. 2007, Fouré et al. 2009, 2010, Houghton et al. 2013). The authors should be more precise at this point.

Line 194ff: It seems interesting that the transverse stiffness and the transverse strain decrease. Can the authors comment on that?

Author Response

(The authors gave the same response as above.)
